## [Peer Review File · The EMBO Journal]

Direct visualization and tracing of chromatin folding in the *Drosophila* embryo

Fadwa Fatmaoui, Pacal Carrivain, Fatima Taiki, Amina Iusupova, Diana Grewe, Wim Hagen, Burkhard Jakob, Jean-Marc Victor, Amélie Leforestier, and Mikhail Eltsov

Corresponding author(s): Mikhail Eltsov (eltsovm@igbmc.fr) , Amélie Leforestier (amelie.leforestier@universite-paris-saclay.fr), Jean-Marc Victor (jean-marc.victor@sorbonne-universite.fr)

Review Timeline:

Submission Date:	3rd Apr 25
Editorial Decision:	22nd May 25
Revision Received:	13th Aug 25
Editorial Decision:	16th Oct 25
Revision Received:	24th Nov 25
Accepted:	11th Dec 25

Editor: Cornelius Schneider

Transaction Report:

Dear Prof. Eltsov,

Thank you for submitting your manuscript for consideration by the EMBO Journal. It has now been seen by three referees whose comments are shown below.

Given the referees' positive recommendations, I would like to invite you to submit a revised version of the manuscript, addressing the comments of all three reviewers. I should add that it is EMBO Journal policy to allow only a single round of revision, and acceptance of your manuscript will therefore depend on the completeness of your responses in this revised version.

Thank you for the opportunity to consider your work for publication. I look forward to your revision.

Yours sincerely,

Cornelius Schneider, PhD
Editor
The EMBO Journal
c.schneider@embojournal.org

- a point-by-point response to the referees' comments, with a detailed description of the changes made (as a word file).
 - a word file of the manuscript text.
 - individual production quality figure files (one file per figure)
 - a complete author checklist, which you can download from our author guidelines (<https://www.embopress.org/page/journal/14602075/authorguide>).
 - Expanded View files (replacing Supplementary Information)
- Please see out instructions to authors
<https://www.embopress.org/page/journal/14602075/authorguide#expandedview>
- a Reagents and Tools Table as part of the Methods section, which can be downloaded from our author guidelines (<https://www.embopress.org/page/journal/14602075/authorguide#structuredmethods>)

We realize that it is difficult to revise to a specific deadline. In the interest of protecting the conceptual advance provided by the work, we recommend a revision within 3 months (20th Aug 2025). Please discuss the revision progress ahead of this time with the editor if you require more time to complete the revisions. Use the link below to submit your revision:

Referee #1:

This chromatin cryo-ET study uses frozen-hydrated *Drosophila* embryos as a starting material, which is a substantial step up from previous ones, which use cultured cells. Owing to the difficulties in cryo-ET sample prep of "thick" samples like tissues and small metazoans, Fatmaoui et al. use high-pressure freezing and cryosectioning. In their cryotomograms, densities that resemble linker DNA are present. Using an approach similar to Beel et al. 2021, the authors traced the path of the chromatin in their denoised tomograms by observing linker DNA densities in between nucleosomes. The authors also observed densities in their tomograms that they claim to resemble non-canonical nucleosome species. Overall and in combination with previous work, this study supports a model in which nucleosomes in eukaryotes follow an irregular zig-zag path, possibly interspersed with non-canonical nucleosomes. Therefore, the conclusions are non-controversial and potentially add new structural insights into chromatin's in situ structure. The data are beautiful and the efforts to produce and analyze them must have been Herculean, but the overall story needs to be tempered by realistic assessment of resolution.

Major issues:

The paper could be greatly improved if the key claims are tempered, especially in the abstract and introduction, so that readers understand that in the absence of subtomogram averaging, the state of the art is insufficient to distinguish between nucleosome models that differ in their histone content.

While denoising tools have been very helpful in helping visualizing densities in tomograms; and it could well be correct that the densities indicated by the authors did indeed correspond to non-canonical nucleosomes, I think more rigorous evidence is needed to justify the identification of these densities (for example, a higher-resolution subtomogram average of the non-canonical nucleosome or a cryo-EM tag that somehow recognizes these non-canonical nucleosomes, though I don't see how the latter is achievable).

Page 5/6 & Fig. S3: The reported resolution for the subtomogram averages was $\sim 13\text{\AA}$ but the STA density shown in Fig. S3 looks lower in resolution. This discrepancy needs to be resolved because the claims about non-canonical nucleosomes hinges on an accurate - and realistic - assessment of the resolution.

Minor issues:

Title: "Chromatin fold" is more accurate because this paper is about chromatin, not just DNA.

Abstract:

DNA in situ in intact flash-frozen organism → DNA in situ from a flash-frozen organism

"non-octameric particles were observed as individual objects, without structure averaging, highlighting the high structural heterogeneity of native chromatin."

This statement needs to be toned down because it implies with a high degree of confidence that the observed structures are non-octameric even though the resolution of the data is insufficient to support this claim.

P4: "has revealed nucleosome structure in situ" please add some more recent studies:

Kreysing et al.: <https://www.biorxiv.org/content/10.1101/2025.04.09.647790v1>

Kelley et al.: <https://www.biorxiv.org/content/10.1101/2024.12.28.630444v1>

Please note that the subtomogram averages in these papers also have biological details that support the claimed resolutions.

P5: "accommodating different DNA and histone amounts, including non-octameric structures". Again, the data are insufficient to support these claims.

P5: "euchromatin and facultative heterochromatin"

Please describe how the fHC and ECfHC were localized in the cryosections.

P6: "the density pattern visible in many top view particles is very similar to projection of α -helix histone fold of the octamer"
This sentence is claiming to distinguish secondary structures $\sim 6\text{\AA}$ wide in an unaveraged tomographic slice. The conventional criteria for analysis of secondary structure is at the bare minimum 10\AA resolution, which is not convincing from the images shown.

P7: "nucleosome structure at 13.6\AA resolution in ECfHC1 and 2, and 12.9\AA in cHC"

The claimed resolution is not supported by the appearance of the density maps. At this resolution, DNA major grooves should be visible. We know this from the tens of ribosome single-particle reconstructions in the 2000s that were done at resolutions similar to the ones claimed here. The density maps here are more in line with the $\sim 25 - 30\text{\AA}$ resolution that others in the field have achieved.

P10: "simulated at the resolution of our data"

If the simulations were done at $\sim 13\text{\AA}$, then this needs to be redone at a more realistic resolution between 25 and 30\AA , congruent with the features visible in the subtomogram averages.

P11: "let us to describe various breathing modes"

Cryo-ET cannot distinguish between snapshots of a dynamic process versus stable, but distorted structures. This claim would be better substantiated if the data showed multiple intermediate conformational states expected of breathing, at sufficient resolution to justify the identification of each intermediate state.

P11: "Their small number, combined with their structural diversity, prevents sub-tomogram averaging approaches"

It is not clear how many sub-nucleosomal particles in total the authors picked out in their tomograms. Please state or put in table how many particles of each sub-nucleosomal species was picked out.

Fig. 1A: It would be helpful to annotate the nuclear envelope.

Fig. 1A: The high-density region in the nucleus is indicated to be the constitutive heterochromatin domain ("cHC"). In this example (Fig. 1A), the low mag image is from a freeze-substituted sample. It is not clear where this high-density region is in the low mag image of the vitreous section (for example, in Fig. S1E). The authors should describe how they locate the cHC region in their vitreous sections, preferably an example image.

Following the point above, if the cHC is not readily distinguishable in their low-mag image of the vitreous sections, like in their freeze substituted sample, how do the authors know that the tilt series they collected of a specific region corresponds to the cHC and not an fHC (facultative heterochromatin)?

Fig. 1K: Please show the corresponding tomogram slice of the isosurface render.

Fig. 1: What are the gray boxes in Fig 1H, I, and K?

Fig S5: Panels C & D - please specify in the caption that "GT" means ground truth.

Fig. S2C, it would be easier to compare the cryo-EM central slice if the contrast were inverted, like in the tomogram, and if it has the same "thickness" as the tomographic slice.

Fig S3: What are the small gray boxes in the Fig. S3 FSC plots?

Fig. S5A: The cartoon under "straight linker" is missing.

Lu Gan

Referee #2:

The authors conducted in situ chromatin structural analysis using thin cryo-sections of the central nervous system from *Drosophila* embryos. By applying Volta Phase Plate (VPP) cryo-electron tomography in combination with deep learning-based denoising, they successfully identified nucleosome particles and traced linker DNA within chromatin. Comparison of their in situ structural data with classical solenoid and zig-zag models suggests that zig-zag conformations are predominant in vivo. They also identified several atypical sub-nucleosomal particles, including hexasomes and hemisomes, as well as a particle resembling an overlapping dinucleosome, which had previously been characterized only in vitro. This suggests, for the first time, their presence in situ.

This study presents a technically ambitious and impactful visualization of chromatin architecture by quantitatively comparing linker DNA length and curvature across regions of constitutive heterochromatin, early compacted facultative heterochromatin, and euchromatin. These findings make a significant contribution to our understanding of in situ chromatin organization. In particular, the observation of atypical sub-nucleosomal particles in their native cellular context is especially intriguing and is likely to attract strong interest from researchers in the field.

However, several concerns remain regarding the accuracy of nucleosome picking and the integrality of linker DNA tracing. The authors should clearly specify the coverage range achieved in their analysis of chromatin architecture.

Major comments

Comment 1:

To evaluate the proportion and accuracy of manually picked nucleosomes, the authors should include a figure showing the picking coordinates of a representative tomogram. Additionally, a comprehensive figure showing the linker DNA segments successfully traced within the tomograms should also be provided.

Comment 2:

The results shown in Figure 3G are not described in the Results section. The authors must refer to and explain the findings of this figure in the main text.

Comment 3:

The in vivo observation of putative sub-nucleosomal particles is fascinating, but it also needs cautious interpretation. Given that histone cores typically exhibit relatively low contrast in electron microscopy, it is possible that the denoising process may have led to the loss of histone-derived densities. This point should be explicitly discussed in the main text.

Comment 4:

The authors should include a schematic description of the subtomogram averaging workflow.

Comment 5:

The authors state that "Besides nucleosomes, other molecular complexes such as chaperonins and proteasomes are readily recognized in the nucleoplasm (Supplementary Figure S2C)." However, given the limited resolution of the tomograms, I suppose it is difficult to definitively identify the densities observed as chaperonins or proteasomes. It would be desirable either to back up these attributions with template matching and subtomogram averaging showing model fitting, or to revise the statement to avoid definitive assertions.

Comment 6:

I would like to know if there are any specific localization patterns or positional characteristics observed for the hexasomes, hemisomes, or overlapping dinucleosomes, observed by the authors. Clarifying their spatial distribution within the nucleus could enhance the overall impact of the manuscript.

Minor comments

Comment 1:

Extra frame lines are present in the schematic illustrations inserted in Figures 1 and 4. Please verify that these lines are removed in the high-resolution versions of the figures.

Comment 2:

Figure 4D: The rotation angles applied to the maps should be indicated.

Comment 3:

Figure 4E: The layout of the maps could be improved. In particular, the position of the indicators showing map orientation is quite confusing. I recommend the authors to reorganize the figure to clarify spatial relationships.

Comment 4:

Supplementary Figure S3:

Please indicate the rotation direction for each map. In addition, the FSC curve graphs appear disordered; please review the formatting to ensure a clear presentation.

Comment 5:

Supplementary Figure S5:

The nucleosome model corresponding to the straight linker appears to be missing. Please correct this.

Referee #3:

Chromatin organization plays a critical role in various genome functions, including transcription and DNA replication/repair. Fatmaoui et al. determined the in situ configurations of nucleosomes in the intact CNS of *Drosophila* embryos using cutting-edge VPP cryo-ET coupled with deep learning-based denoising. The authors followed DNA trajectories in situ in euchromatin and facultative heterochromatin regions. The DNA path was visualized as it wrapped around and between nucleosome particles, revealing both local chromatin folding and nucleosome conformations at the individual level. They directly traced a zig-zag folding path over 3 to 4 successive nucleosomes, uncovering a disordered zig-zag pattern. In addition, they observed nucleosome conformational variability, including non-canonical structures and non-octameric particles. This study was very carefully conducted. The findings are impressive and highly impactful for the chromatin field. I support its publication. For The EMBO Journal, with its broad readership, my specific comments are as follows:

Major points:

- 1) For general readers who may not be familiar with cryo-EM, it would be helpful to list the processing algorithms used (e.g., NAD, Warp, Topaz etc.) in a table, along with brief explanations. In addition, providing a list of abbreviations (e.g., CNS, cHC, NO, EC, fHC, STA, HPF, VPP, etc.) would improve the clarity of the paper.
- 2) Although the authors focused on the detailed arrangements of nucleosomes in situ, it is well known that chromatin structure is not static, and nucleosomes are highly mobile (i.e., fluctuating like a liquid) in live cells—even in heterochromatin (e.g., PMID: 32574554; 37018405; 40153514; 40301047). Such dynamics result in variable nucleosome conformations. This dynamic property directly influences nucleosome fiber structure and genome functions. In the Introduction and Discussion sections, incorporating descriptions of the liquid-like aspects of nucleosomes would make the paper more appealing to general readers.
- 3) The authors mainly focused on ECfHC regions, which include euchromatin (EC) and facultative heterochromatin (fHC), but these two might be structurally distinct. Please clarify this point.
- 4) The authors may also consider discussing a recent relevant study (Kreysing et al., bioRxiv 2025; <https://doi.org/10.1101/2025.04.09.647790>), which could provide additional context.
- 5) Page 11. "All these non-canonical nucleosome-related structures were only found in ECfHC nanodomains, corresponding to about 2% of the nucleosome population." This finding is very impressive. I wonder whether the authors could estimate how many genes (or what fraction) are expressed in the analyzed cells.

Minor points:

- 1) Page 3. I am not sure that high-resolution Hi-C can reveal local zig-zag or solenoidal folding (Ohno et al., 2019; Risca et al., 2017).
- 2) Page 4. "...with modifications of the osmotic and ionic environment..." Please note that such changes can affect chromatin structure.
- 3) Page 4. "The addition of glycerol has also been used (Chen et al., 2025; Hou et al., 2023)..." Note that Chen et al. mainly used DMSO.
- 4) Page 6. "The network training was successful only for reconstructions with <0.6 nm of the mean residual error measured in etomo (Mastronarde & Held, 2017)." For smoother flow, it might be better to move this sentence to the Methods section.
- 5) Page 12. "...for example, bound linker histones resulting in the closed conformation of a chromatosome (Hayes et al., 1994)..." Transient binding of linker histones was recently demonstrated (Shimazoe et al., bioRxiv 2025; <https://doi.org/10.1101/2025.03.05.641622>).

Manuscript EMBOJ-2025-120942
Answer to the Referees

Referee #1:

This chromatin cryo-ET study uses frozen-hydrated *Drosophila* embryos as a starting material, which is a substantial step up from previous ones, which use cultured cells. Owing to the difficulties in cryo-ET sample prep of "thick" samples like tissues and small metazoans, Fatmaoui et al. use high-pressure freezing and cryosectioning. In their cryotomograms, densities that resemble linker DNA are present. Using an approach similar to Beel et al. 2021, the authors traced the path of the chromatin in their denoised tomograms by observing linker DNA densities in between nucleosomes. The authors also observed densities in their tomograms that they claim to resemble non-canonical nucleosome species. Overall and in combination with previous work, this study supports a model in which nucleosomes in eukaryotes follow an irregular zig-zag path, possibly interspersed with non-canonical nucleosomes. Therefore, the conclusions are non-controversial and potentially add new structural insights into chromatin's in situ structure. The data are beautiful and the efforts to produce and analyze them must have been Herculean, but the overall story needs to be tempered by realistic assessment of resolution.

Major issues:

The paper could be greatly improved if the key claims are tempered, especially in the abstract and introduction, so that readers understand that in the absence of subtomogram averaging, the state of the art is insufficient to distinguish between nucleosome models that differ in their histone content.

We agree with the Referee, the resolution of our data is not sufficient to demonstrate histone contents in the different particles. Our claims should indeed be taken as hypotheses, based on differences of DNA content and core density patterns. We therefore modified the abstract, introduction and discussion to avoid overstatements.

While denoising tools have been very helpful in helping visualizing densities in tomograms; and it could well be correct that the densities indicated by the authors did indeed correspond to non-canonical nucleosomes, I think more rigorous evidence is needed to justify the identification of these densities (for example, a higher-resolution subtomogram average of the non-canonical nucleosome or a cryo-EM tag that somehow recognizes these non-canonical nucleosomes, though I don't see how the latter is achievable).

We agree with the Referee that high resolution STA would be needed for a full proof of evidence of sub-nucleosome particles and in particular the distinction between hemi- and tetra-some. This is at the moment not realistic. These particles represent about 1 to 2 % of the total nucleosome population we identified and manually picked in our ECfHC tomograms, i.e. 5 to 13 per tomogram, and our data suggest that they include different types of subnucleosomes. There are probably a bit more than what we detected, but, to reach such a goal one would not only need thousands of tomograms, but also to develop automated methods to search reliably for them, an approach which is already not perfect for canonical nucleosomes. Although cryo-EM tags were recently designed to target histone variants (doi: [10.1101/2024.10.12.617288](https://doi.org/10.1101/2024.10.12.617288)), this approach cannot be used to detect tetra and hemisomes that are probably very transient species that may not differ chemically from their octameric neighbors. Additionally, single gyre particles are expected to be transient dynamic entities with various conformational states, possibly also affected by additional binders. We foresee

that STA or labelling approaches has to be established in vitro before going to in situ experiments.

Page 5/6 & Fig. S3: The reported resolution for the subtomogram averages was ~13Å but the STA density shown in Fig. S3 looks lower in resolution. This discrepancy needs to be resolved because the claims about non-canonical nucleosomes hinges on an accurate - and realistic - assessment of the resolution.

We are very thankful to the referee for pointing out this issue. We examined our averaging experiments and found that the Artiatomi software version used in the previous submission does not provide the gold standard validation by default. We have redone the STA part completely and performed gold standard averaging for the total particle set (~2000 particles) and the gold standard (GS) resolution assessment indeed indicates ~20 Å resolution. This was the maximum that we could obtain after exploration of different settings and masks for the alignment. The processing workflow and best setting results are added to the Expanded view (EV2) and Material and Methods section. The final average, and even and odd averages are provided to the referees and will be uploaded to EMDB.

Due to a limited number of nucleosomes picked per tomogram (~500 to ~800) we decided to not claim the STA resolution for individual tomograms, and, in this case, apply STA without GS splitting, only for the purpose mapping back nucleosomes in tomograms. It showed generally a good correspondence of nucleosome positions and orientations with what can be identified visually.

Minor issues:

Title: "Chromatin fold" is more accurate because this paper is about chromatin, not just DNA. We modified the title as suggested.

Abstract:

DNA in situ in intact flash-frozen organism → DNA in situ from a flash-frozen organism

"non-octameric particles were observed as individual objects, without structure averaging, highlighting the high structural heterogeneity of native chromatin."

This statement needs to be toned down because it implies with a high degree of confidence that the observed structures are non-octameric even though the resolution of the data is insufficient to support this claim.

The abstract has been modified as suggested.

P4: "has revealed nucleosome structure in situ" please add some more recent studies: Kreysing et al.: <https://www.biorxiv.org/content/10.1101/2025.04.09.647790v1> Kelley et al.: <https://www.biorxiv.org/content/10.1101/2024.12.28.630444v1>

Please note that the subtomogram averages in these papers also have biological details that support the claimed resolutions.

We added these two new references.

P5: "accommodating different DNA and histone amounts, including non-octameric structures". Again, the data are insufficient to support these claims.

We modified the sentence as: "We also detected individually a variety of nucleosome

conformations, among which particles accommodating different DNA amounts wrapped

around, from less than one to three gyres, localized in ECfHC nanodomains.”

P5: "euchromatin and facultative heterochromatin"

Please describe how the fHC and ECfHC were localized in the cryosections.

Our identification is based on the results a correlative light and electron microscopy performed on freeze-substituted *Drosophila* embryos expressing HP1a-GFP construct. This experiment let us demonstrate unambiguously that large cHC domains are located close to the nuclear envelope and nucleolus, whereas EC and fHC are dispersed within the nucleoplasm as multiple nano-domains. Such reproducible general organization of embryonic brain nuclei enabled identification of cHC versus ECfHC domains in the cryo-section without CLEM, on the basis of the localization of chromatin domains in relation to the structural marks like nuclear envelope and nucleolus. This is now mentioned in the introduction, and described in Materials and Methods and Expanded view Figure EV1, and Appendix Figure S1

P6: "the density pattern visible in many top view particles is very similar to projection of a-helix histone fold of the octamer"

This sentence is claiming to distinguish secondary structures $\sim 6\text{\AA}$ wide in an unaveraged tomographic slice. The conventional criteria for analysis of secondary structure is at the bare minimum 10\AA resolution, which is not convincing from the images shown.

Thank you for noticing. It is not straightforward to determine the resolution of cryo-tomograms, but we agree that our formulation may be misleading to an overstatement. We modified the sentence as follows: "In addition, the density pattern visible in many top view particles is very similar to the that of the core histone, with its typical symmetrical M- shape, and clear identification of the dyad axis, which is confirmed by the path of the DNA". We also added to Figure 1 a virtual slice of a simulated tomogram of the 2PYO nucleosome at 25\AA resolution, to be compared with our experimental pattern.

P7: "nucleosome structure at 13.6\AA resolution in ECfHC1 and 2, and 12.9\AA in cHC" The claimed resolution is not supported by the appearance of the density maps. At this resolution, DNA major grooves should be visible. We know this from the tens of ribosome single-particle reconstructions in the 2000s that were done at resolutions similar to the ones claimed here. The density maps here are more in line with the $\sim 25 - 30\text{\AA}$ resolution that others in the field have achieved.

We fully agree with the referee and made modifications in the corresponding part of the text and figures.

P10: "simulated at the resolution of our data"

If the simulations were done at $\sim 13\text{\AA}$, then this needs to be redone at a more realistic resolution between 25 and 30\AA , congruent with the features visible in the subtomogram averages.

We implemented the requested changes and modified the resolution of simulations accordingly. New simulations were done at 25\AA . They are to be compared with individual patterns. These are selected as the best, unambiguous situations where the DNA curvature can be measured, and the typical core histone pattern is visualized. There, the local resolution can be expected higher than the 18\AA of our STAs. We now indicate in the text the resolution used in the simulations.

P11: "let us to describe various breathing modes"

Cryo-ET cannot distinguish between snapshots of a dynamic process versus stable, but

distorted structures. This claim would be better substantiated if the data showed multiple

intermediate conformational states expected of breathing, at sufficient resolution to justify the identification of each intermediate state.

Thank you for this comment. Indeed, we did not quantify the opening angle of DNA at the entry/exit site of the nucleosome, which would be necessary to support the breathing hypothesis. We modified the discussion accordingly.

P11: "Their small number, combined with their structural diversity, prevents sub-tomogram averaging approaches"

It is not clear how many sub-nucleosomal particles in total the authors picked out in their tomograms. Please state or put in table how many particles of each sub-nucleosomal species was picked out.

We detailed in the following sentence the number of manually picked nucleosomes and sub-nucleosomes in the analyzed tomograms: "In two of our ECfHC tomograms, we manually picked 549 and 552 nucleosomes (used for STA), and identified 8 and 13 1-gyre particles respectively, which corresponds to 1.5 - 2.4 % of the nucleosome population. They appear dispersed within or at the periphery of ECfHC domains, in some cases as pairs of particles. Appendix Figure S8 shows their repartitions within the two ECfHC tomograms used for STA. In this work, we additionally explored 4 ECfHC tomograms, where a few (5 to 10) 1-gyre particles were found in each of them. Note that the number of both picked nucleosomes and sub-nucleosomes per tomogram is underestimated, as only unambiguously recognized particles were considered."

We also report in a new Appendix Figure S8 their location among picked nucleosomes of the two tomograms used for STA of ECfHC. Note that the number of both picked nucleosomes and sub-nucleosomes per tomogram is likely underestimated, as only unambiguously recognized particles were considered.

We also perform a brief screening of nucleosome looking densities in ECfHC3 and a few other ECfHC-containing reconstructions (not illustrated), and could identify and locate there several sub-nucleosomal particles per ECfHC tomogram. However, the proper quantitative assessment of their ratio in relation to the nucleosomes cannot be achieved with such a manual picking approach, because the number of both picked nucleosomes and subnucleosomes per tomogram is underestimated. We fully agree that it would be a very important information, but in the present state it is beyond our reach.

Fig. 1A: It would be helpful to annotate the nuclear envelope. **This has been done.**

Fig. 1A: The high-density region in the nucleus is indicated to be the constitutive heterochromatin domain ("cHC"). In this example (Fig. 1A), the low mag image is from a freeze-substituted sample. It is not clear where this high-density region is in the low mag image of the vitreous section (for example, in Fig. S1E). The authors should describe how they locate the cHC region in their vitreous sections, preferably an example image.

Example images have been added in the new Appendix Figure S1. Wherever nuclear envelope can be identified in low-magnification images (used as Anchor images in Serial EM), we could target the tilt series acquisition using this structural feature: the cHC regions were identified in tilt series targeted to the nuclear periphery, whereas ECfHC domains were targeted close to the center of the nucleus.

Following the point above, if the cHC is not readily distinguishable in their low-mag image of the vitreous sections, like in their freeze substituted sample, how do the authors know that the

tilt series they collected of a specific region corresponds to the cHC and not an fHC (facultative heterochromatin)?

As we now show by CLEM of freeze-substituted samples in the new Expanded view Figure EV1, the large compact chromatin domains close to the nuclear envelope and the nucleolus correspond to cHC. In contrast, EC and fHC form nanodomains dispersed within the nucleoplasm. In cryo-sections, large compact domains found at the vicinity of the nuclear envelope and/or nucleolus are therefore unambiguously always cHC. Small (and looser) clusters found far (> 500 nm) away from a well identified nuclear envelope are ECfHC nanodomains. Of course, in the absence of cryo-CLEM identification, there are some situations where the characterization of the chromatin compartment cannot be established unambiguously. We did not consider these cases in our analyses here (STA, linker tracing).

Fig. 1K: Please show the corresponding tomogram slice of the isosurface render. **This has been added**

Fig. 1: What are the gray boxes in Fig 1H, I, and K?

These were artefacts from pdf generation and are absent in final figure

Fig S5: Panels C & D - please specify in the caption that "GT" means ground truth. **This has been added**

Fig. S2C, it would be easier to compare the cryo-EM central slice if the contrast were inverted, like in the tomogram, and if it has the same "thickness" as the tomographic slice. **This has been done. The cryo-EM densities now shown were simulated at the resolution of 25 Å.**

Fig S3: What are the small gray boxes in the Fig. S3 FSC plots?

These were artefacts from pdf generation and are absent in final figure

Fig. S5A: The cartoon under "straight linker" is missing.

This has been corrected.

Lu Gan

Referee #2:

The authors conducted in situ chromatin structural analysis using thin cryo-sections of the central nervous system from *Drosophila* embryos. By applying Volta Phase Plate (VPP) cryo-electron tomography in combination with deep learning-based denoising, they successfully identified nucleosome particles and traced linker DNA within chromatin. Comparison of their in situ structural data with classical solenoid and zig-zag models suggests that zig-zag conformations are predominant in vivo.

They also identified several atypical sub-nucleosomal particles, including hexasomes and hemisomes, as well as a particle resembling an overlapping dinucleosome, which had previously been characterized only in vitro. This suggests, for the first time, their presence in situ.

This study presents a technically ambitious and impactful visualization of chromatin architecture by quantitatively comparing linker DNA length and curvature across regions of constitutive heterochromatin, early compacted facultative heterochromatin, and euchromatin.

These findings make a significant contribution to our understanding of in situ chromatin organization. In particular, the observation of atypical sub-nucleosomal particles in their native cellular context is especially intriguing and is likely to attract strong interest from

researchers in the field.

However, several concerns remain regarding the accuracy of nucleosome picking and the integrality of linker DNA tracing. The authors should clearly specify the coverage range achieved in their analysis of chromatin architecture.

Major comments

Comment 1:

To evaluate the proportion and accuracy of manually picked nucleosomes, the authors should include a figure showing the picking coordinates of a representative tomogram. Additionally, a comprehensive figure showing the linker DNA segments successfully traced within the tomograms should also be provided.

The new Expanded view Figure EV5 and Appendix Figure S9 show an example of a tomogram with STA mapped back together with traced linkers. Note that linkers shown are those where the two datasets obtained independently - linkers on the one hand, picked particles on the other hand - intercept. Appendix Figure S9 is also shown in the form of a new Supplementary movie S5.

Comment 2:

The results shown in Figure 3G are not described in the Results section. The authors must refer to and explain the findings of this figure in the main text.

We added a description of Figure 3G.

Comment 3:

The in vivo observation of putative sub-nucleosomal particles is fascinating, but it also needs cautious interpretation. Given that histone cores typically exhibit relatively low contrast in electron microscopy, it is possible that the denoising process may have led to the loss of histone-derived densities. This point should be explicitly discussed in the main text.

Indeed, histone core show low density patterns that could possibly be modified by deep learning denoising (note however that denoised nucleosomes in favorable orientation and local environment, such as those shown in Figure 1, do show the density aspect expected from the cryo-EM density of histone octamer imaged at the corresponding resolution). But the number of DNA gyres wrapped around can be unambiguously visualized already in raw tomograms (and more clearly in simple nad-filtered tomograms), demonstrating that the visualization of these particles does not result from deep-learning denoising. This is now shown in Appendix Figure S5 (for the example of the 1-gyre particle shown in Expanded view EV4).

Comment 4:

The authors should include a schematic description of the subtomogram averaging workflow.

This is now added in Expanded view Figure EV2.

Comment 5:

The authors state that "Besides nucleosomes, other molecular complexes such as chaperonins and proteasomes are readily recognized in the nucleoplasm (Supplementary Figure S2C)." However, given the limited resolution of the tomograms, I suppose it is difficult to definitively identify the densities observed as chaperonins or proteasomes. It would be

desirable either to back up these attributions with template matching and subtomogram averaging showing model fitting, or to revise the statement to avoid definitive assertions.

Proteasome and chaperonins have very characteristic size and shape that can be readily identified. But we did not run template matching and STA of them in our tomograms. First, we expect only a very limited number of targeted particles (< 50). The template matching in

such situation will very unlikely serve as the validation tool because of the risk of the template bias. However, we accepted the concern of the referee and reformulated our statements into a more hypothetical tone. We updated the Supplementary Figure S2C (now Appendix Figure S2C) according to the suggestion of the referee #1, showing simulated tomographic slices from PDB structures at 25 Å resolution, so that these comparative figures have the aspect and resolution of the cryo-EM densities visible in our data.

Comment 6:

I would like to know if there are any specific localization patterns or positional characteristics observed for the hexasomes, hemisomes, or overlapping dinucleosomes, observed by the authors. Clarifying their spatial distribution within the nucleus could enhance the overall impact of the manuscript.

These particles are found dispersed within the ECfHC nanodomains. Many are found at the periphery of these domains, but not all. Two examples are now shown in Appendix Figure S8.

Minor comments

Comment 1:

Extra frame lines are present in the schematic illustrations inserted in Figures 1 and 4. Please verify that these lines are removed in the high-resolution versions of the figures.

We checked that these lines are removed.

Comment 2:

Figure 4D: The rotation angles applied to the maps should be indicated. The rotation angles have been added.

Comment 3:

Figure 4E: The layout of the maps could be improved. In particular, the position of the indicators showing map orientation is quite confusing. I recommend the authors to reorganize the figure to clarify spatial relationships.

We did our best to improve the orientation indicators in Figure 4C & E, in particular with a sketch of the sub-volume shown in Figure 4E. Additionally, most of the maps shown in the figure are also provided as Supplementary movies showing the complete rotation range around the z-axis.

Comment 4:

Supplementary Figure S3:

Please indicate the rotation direction for each map. In addition, the FSC curve graphs appear disordered; please review the formatting to ensure a clear presentation.

The figure has been entirely reviewed with new STAs and FSC curves (now Expanded view EV2). We took care to add the rotation director.

Comment 5:

Supplementary Figure S5:

The nucleosome model corresponding to the straight linker appears to be missing. Please correct this.

This has been corrected.

Referee #3:

Chromatin organization plays a critical role in various genome functions, including transcription and DNA replication/repair. Fatmaoui et al. determined the in situ configurations of nucleosomes in the intact CNS of *Drosophila* embryos using cutting-edge VPP cryo-ET coupled with deep learning-based denoising. The authors followed DNA trajectories in situ in euchromatin and facultative heterochromatin regions. The DNA path was visualized as it wrapped around and between nucleosome particles, revealing both local chromatin folding and nucleosome conformations at the individual level. They directly traced a zig-zag folding path over 3 to 4 successive nucleosomes, uncovering a disordered zig-zag pattern. In addition, they observed nucleosome conformational variability, including non-canonical structures and non-octameric particles. This study was very carefully conducted. The findings are impressive and highly impactful for the chromatin field. I support its publication. For The EMBO Journal, with its broad readership, my specific comments are as follows:

Major points:

1) For general readers who may not be familiar with cryo-EM, it would be helpful to list the processing algorithms used (e.g., NAD, Warp, Topaz etc.) in a table, along with brief explanations. In addition, providing a list of abbreviations (e.g., CNS, cHC, NO, EC, fHC, STA, HPF, VPP, etc.) would improve the clarity of the paper.

The list of processing algorithms was added to the Reagents and Tools tables (submitted as a separate file), and technical comments were moved to Appendix Figure S2. A list of abbreviations is now provided at the end of the manuscript.

2) Although the authors focused on the detailed arrangements of nucleosomes in situ, it is well known that chromatin structure is not static, and nucleosomes are highly mobile (i.e., fluctuating like a liquid) in live cells—even in heterochromatin (e.g., PMID: 32574554; 37018405; 40153514; 40301047). Such dynamics result in variable nucleosome conformations. This dynamic property directly influences nucleosome fiber structure and genome functions. In the Introduction and Discussion sections, incorporating descriptions of the liquid-like aspects of nucleosomes would make the paper more appealing to general readers.

We agree that our observations support the liquid like dynamics of chromatin. We also fully agree about the importance of these relatively recent observations for understanding of the functional chromatin organization. However, to more quantitatively explore this point, more data and measurements (including for example nearest neighbor distribution function, and DNA entry-exit angle distribution) would be required. It is therefore at the moment too early to properly discuss this aspect. We nonetheless introduced the notion of liquid behavior in the introduction and discussion so that readers are aware of this link.

3) The authors mainly focused on ECfHC regions, which include euchromatin (EC) and facultative heterochromatin (fHC), but these two might be structurally distinct. Please clarify this point.

We would be very fascinated to have a labelling system that would enable distinguishing these two types of chromatin in *Drosophila* embryos. At the moment, we have no way to distinguish between EC and fHC regions in our cryo-tomograms. This would require further investigation using complex cryo-CLEM approaches. We modified the description p6, so that this point is clear to readers.

4) The authors may also consider discussing a recent relevant study (Kreysing et al., bioRxiv 2025; <https://doi.org/10.1101/2025.04.09.647790>), which could provide additional context.

This reference is now added.

5) Page 11. "All these non-canonical nucleosome-related structures were only found in ECfHC nanodomains, corresponding to about 2% of the nucleosome population." This finding is very impressive. I wonder whether the authors could estimate how many genes (or what fraction) are expressed in the analyzed cells.

There is still a long way to go to answer this question. Our data are obtained on ultrathin sections, each corresponding to less than 0.1 % of the nucleus volume. In addition, the number of particles varies from one ECfHC region to another (from 5 to 13 particles detected per tomogram), which may also reflect different activities. This aspect is reinforced by the fact that we do not distinguish EC from fHC domains. Which means that the link between these particles and gene expression is not demonstrated, only hypothesized. Lastly, the number of detected particles here is only an estimation (and, as mentioned, probably underestimated). Beyond the hypothesis of a link between these particles and gene expression evoked in the manuscript, further discussion, including the number of expressed genes, seems too speculative at the moment.

Minor points:

1) Page 3. I am not sure that high-resolution Hi-C can reveal local zig-zag or solenoidal folding (Ohno et al., 2019; Risca et al., 2017).

Hi-C data from these references were shown to be compatible with - rather than demonstrate – these local folds. The sentence has been modified.

2) Page 4. "...with modifications of the osmotic and ionic environment..." Please note that such changes can affect chromatin structure.

This is now clearly mentioned in the text.

3) Page 4. "The addition of glycerol has also been used (Chen et al., 2025; Hou et al., 2023)..." Note that Chen et al. mainly used DMSO.

This has been added.

4) Page 6. "The network training was successful only for reconstructions with <0.6 nm of the mean residual error measured in etomo (Mastrorarde & Held, 2017)." For smoother flow, it might be better to move this sentence to the Methods section.

The sentence has been moved to the Material & Method section.

5) Page 12. "...for example, bound linker histones resulting in the closed conformation of a chromatosome (Hayes et al., 1994)..." Transient binding of linker histones was recently demonstrated (Shimazoe et al., bioRxiv 2025; <https://doi.org/10.1101/2025.03.05.641622>).

This aspect has been included in the discussion.

Dear Dr Etsov,

Thank you for submitting a revised version of your manuscript. Your study has now been seen by all original referees. Referees #2 and 3 find that their previous concerns have been addressed and now recommend publication of the manuscript. Referee #1 also agrees with your response to the concerns raised during the first round of revision however thinks that the identified caveats should be clearly stated in the manuscript including the abstract. I fully agree with the assessment of this referee both regarding the need for the proposed textual edits as well as the statement that this does not detract from the impact of the study and rather illustrates that the manuscript operates at the technical forefront. I would therefore ask you to incorporate the suggestions by referee #1 in the final version of the manuscript. In addition, there remain only a few mainly editorial points that have to be addressed before I can extend formal acceptance of the manuscript:

- FUNDING INFO: "Funding" should be included in "Acknowledgements"; missing info in ms: Agence Nationale de la Recherche (ANR) grant number: ANR-23-CE45-0012-01; missing info in eJP: Centre for Integrative Biology (CBI), CNRS, Inserm, University of Strasbourg; Instruct-ERIC; iNext-Discovery, grant number 871037, funded by the Horizon 2020 program of the European Commission

- On the abstract page of the manuscript, please include 4-5 general keyword terms to enhance searchability.

- Please rename the "Conflict of Interest" section into "Disclosure and Competing Interests Statement", in accordance with our updated Guide to Authors (<https://www.embopress.org/competing-interests>)

- As we are switching from a free-text author contribution statement towards a more formal statement based on Contributor Role Taxonomy (CRediT) terms, please remove the present Author Contribution section and instead specify each author's contribution(s) directly in the Author Information page of our submission system during upload of the final manuscript. See <https://casrai.org/credit/> for more information.

- FIGURE CALLOUTS: all callouts should be listed sequentially; EV figures should be called out as Figure EV1-EV5 instead of Expanded View EV1-EV5 with all the figure panels;

- Please upload EV figures as individual Figure files with legends placed below the main figure legends in ms

- DATASET EV LEGENDS: There is a file "Data SetEV2" upldd as a dataset, but contains 3 videos that are not playing. Please doublecheck.

- APPENDIX 1 FILE WITH ToC: title page should contain "Appendix for + ms title" and ToC with the page numbers for the listed items; nomenclature should be Appendix Figure Sx and Appendix Table Sx throughout ms and Appendix PDF - do not use the word "Supplementary"

- Please provide suggestions for a short 'blurb' text prefacing and summing up the conceptual aspect of the study in two sentences (max. 250 characters), followed by 3-5 one-sentence 'bullet points' with brief factual statements of key results of the paper; they will form the basis of an editor-written 'Synopsis' accompanying the online version of the article. Please also provide an altered synopsis image, making sure that the aspect ratio conforms to our website's format - it should be exactly 550 pixels wide and between 300-600 pixels high.

- There is possible reuse of images between figures in several cases. Please carefully check reuse within the figure set and call out all image reuse in the corresponding figure legends. Possible reuse between:

Figure 1i and Figure 3E

Figure 1G and Figure 4A

Figure 1D and Figure EV4B

Figure EV4 A, A' and Appendix Figure S5 & S7.

Figure EV5A and Appendix Figure S2A

Figure EV5A & B and Appendix Figure S9

Figure Appendix Figure S2B and Appendix Figure S9

- Pixelation in Figure EV and Appendix files. Currently, EV is in one merged file. Please provide High res Individual EV Figures and Appendix file.

- Figure Legends (main + EV):

1. Please note that the box plots need to be defined in terms of minima, maxima, centre, bounds of box and whiskers, and percentile in the legends of figures 3A, B

2. Please note that information related to n is missing in the legends of figures 3A, B

3. Please note that the blue arrows are not defined in the legend of figures 1G, G'.
4. Please note that the dotted borders are not defined in the legend of figures 1B', C'.
5. Please note that the magenta arrow heads are not defined in the legend of figures 1K.
6. Please note that the magenta circles are not defined in the legend of figures 2A, B.
7. Please note that the black and white arrows are not defined in the legend of figures EV1 D, E.

- "MATERIAL AND METHODS" should be renamed to "Methods"

- 5 movie files - playing. They should be renamed to Movie EV1-EV2 with the corresponding callouts, and the legends should be removed from Appendix PDF and zipped with each movie file.

- Sections need to be named and the order should be corrected: Title page - Abstract - Keywords - Introduction - Results - Discussion - Methods - Data Availability - Acknowledgements - Disclosure and Competing Interests Statement - References - Figure Legends - Table(s) - Expanded View Figure Legends.

With best regards,

Cornelius Schneider

Cornelius Schneider, PhD
Editor | The EMBO Journal
c.schneider@embojournal.org

Please refer to our figure preparation guideline in order to ensure proper formatting and readability in print as well as on screen:

See also figure legend guidelines:

<https://www.embopress.org/page/journal/14602075/authorguide#figureformat>

Referee #1:

Major points

The revised manuscript has changes in both new experiments and presentation that make it easier to understand. Furthermore, the rebuttal has more explanations about the limitations of the data and how they should be interpreted. The rebuttal to the minor points all have corresponding changes in the manuscript, but not so for the rebuttal to the first two major points. I strongly urge the authors to move some of these explanations to the text, rather than make the readers have to download the review file for the additional insightful discussion. Readers would benefit from these technical explanations because the results achieved in

this paper are at the limits of the current technology.

"Our claims should indeed be taken as hypotheses, based on differences of DNA content and core density patterns." -- This statement is in the rebuttal and should also be in the abstract. As worded, the abstract sounds too confident that conformationally variable non-canonical structures were observed. The non-expert reader will take in this statement and the multitude of unaveraged cryo-ET densities + docked atomic models, and then conclude that structures like these exist inside cells. I suggested rewording the abstract as follows:

"Additionally, nucleosome conformational variability with non-canonical structures" → "Additionally, **densities consistent with** nucleosome conformational variability with non-canonical structures"

A caveat like this does not detract from the impact of the paper because it provides a more accurate description how one can understand the current state of the art.

Fig EV4 caption: "1 gyre sub-nucleosome particles seen in side view in the XY plane (A') are unambiguously characterized by the visualisation of a single DNA gyre"

"When seen in top view in (or close to) the XY plane, they can be recognized by their characteristic histone pattern. In some of them, the absence of the second tetramer appears as missing densities."

-- and --

Fig S5 caption: "The single DNA gyre is unambiguously recognized in all denoised (Warp/nad) and raw tomograms."

These two captions sound overly confident that the densities they pointed out are actually (non-canonical) nucleosomes. For example, I'm not sure if the "raw" panel in Figure S5 can be considered unambiguously a nucleosome (or a sub-nucleosome), let alone being able to count the number of gyres. These figure captions (and their callouts in the main text) can be made more rigorous if "subnucleosomes/nucleosomes" is replaced with "candidate subnucleosomes/nucleosomes" or "subnucleosome-like densities". Again, the implication of uncertainty (taking the claims as a hypothesis) does not detract from the impact of this paper.

Minor points

Abstract: Here is my attempt to make one of the phrases less awkward:

"less than one to three gyres" → "ranging from less than one up to three gyres"

Fig. 1A - C, 4A: Please flatten the text layers to hide the red spell-check underlines.

Fig. 1G: I suggest adding the label "simulated" to the figure panel so that people who skip the figure legend don't mistake this image for an experimental projection/class average.

Fig 1G/G' and P.6: I don't see the M shape. Is the blue arrow suppose to indicate the M-shaped motif?

Fig EV2: FSC plot -- please add legend.

Fig EV3 caption: "..indicated by the black arrows". -- Did you mean magenta arrows?

Fig EV4: "The thickness of the elongated density varies from about 5 to 8.5 nm for 1-gyre particles, and from about 6.5 to 11 for 2-gyres nucleosomes (determined from their histone pattern in XY plane)" -- Is this text referring to the double-arrow-with-dashed-lines in the figure? If so, please indicate this in the text.

Also, are the solid white arrows in panels A and B' meant to be magenta in color? White arrows are not mentioned in the caption.

-Lu Gan

Referee #2:

The authors have responded as fully as possible to our comments, explicitly addressing the resolution limits and the potential effects of denoising in both the main text and the supplementary materials. Although the limited resolution remains a constraint, the authors have made optimal use of the available data and provided valuable considerations for readers. Therefore, I consider the revised manuscript suitable for acceptance.

Referee #3:

The authors have adequately addressed my comments in the revised manuscript. I am satisfied with their revision. Using cryo-ET, the authors revealed the path of nucleosomal and linker DNA in the crowded cell nucleus. This study is a landmark in cell biology and deserves to be highlighted.

Dear Editor:

We sincerely thank the editor and reviewers for their careful evaluation of our manuscript and for the insightful and encouraging feedback. We are especially grateful to Prof. Lu Gan for the very thorough and constructive comments. We have carefully addressed all points raised and revised the manuscript accordingly. The modifications are highlighted in the revised version in red, and detailed, point-by-point responses are provided below.

Thank you for submitting a revised version of your manuscript. Your study has now been seen by all original referees. Referees #2 and 3 find that their previous concerns have been addressed and now recommend publication of the manuscript. Referee #1 also agrees with your response to the concerns raised during the first round of revision however thinks that the identified caveats should be clearly stated in the manuscript including the abstract. I fully agree with the assessment of this referee both regarding the need for the proposed textual edits as well as the statement that this does not detract from the impact of the study and rather illustrates that the manuscript operates at the technical forefront.

I would therefore ask you to incorporate the suggestions by referee #1 in the final version of the manuscript.

We appreciate this feedback. We performed local text edits to incorporate these suggestions into the abstract, main text, EV, and Appendix. We present our detailed answers under the comments of the reviewer #1.

In addition, there remain only a few mainly editorial points that have to be addressed before I can extend formal acceptance of the manuscript:

- *FUNDING INFO: "Funding" should be included in "Acknowledgements"; missing info in ms: Agence Nationale de la Recherche (ANR) grant number: ANR-23-CE45-0012-01; missing info in eJP: Centre for Integrative Biology (CBI), CNRS, Inserm, University of Strasbourg; Instruct-ERIC; iNext-Discovery, grant number 871037, funded by the Horizon 2020 program of the European Commission*

The funding information has been corrected and included in the "Acknowledgements" section. For the iNEXT-Discovery support, we followed the official citation guidelines of the program regarding access to cryo-EM instrumentation.

- On the abstract page of the manuscript, please include 4-5 general keyword terms to enhance searchability.

Done.

- Please rename the "Conflict of Interest" section into "Disclosure and Competing Interests Statement", in accordance with our updated Guide to Authors (<https://www.embopress.org/competing-interests>)

Done.

- As we are switching from a free-text author contribution statement towards a more formal statement based on Contributor Role Taxonomy (CRediT) terms, please remove the present Author Contribution section and instead specify each author's contribution(s) directly in the Author Information page of our submission system during upload of the final manuscript. See <https://casrai.org/credit/> for more information.

Done.

- FIGURE CALLOUTS: all callouts should be listed sequentially; EV figures should be called out as Figure EV1-EV5 instead of Expanded View EV1-EV5 with all the figure panels;

All EV figure callouts have been checked and updated throughout the text.

- Please upload EV figures as individual Figure files with legends placed below the main figure legends in ms.

Done.

- DATASET EV LEGENDS: There is a file "Data SetEV2" uplidd as a dataset, but contains 3 videos that are not playing. Please doublecheck.

This file was uploaded by mistake in the previous submission. The tomographic data presented in the movies have been deposited in the EMPIAR database under the accession code EMPIAR-12964 and will be released upon publication.

- APPENDIX 1 FILE WITH ToC: title page should contain "Appendix for + ms title" and ToC with the page numbers for the listed items; nomenclature should be Appendix Figure Sx and Appendix Table Sx throughout ms and Appendix PDF - do not use the word "Supplementary"

Done.

- Please provide suggestions for a short 'blurb' text prefacing and summing up the conceptual aspect of the study in two sentences (max. 250 characters), followed by 3-5 one-sentence 'bullet points' with brief factual statements of key results of the paper; they will form the basis of an editor-written 'Synopsis' accompanying the online version of the article. Please also provide an altered synopsis image, making sure that the aspect ratio conforms to our website's format - it should be exactly 550 pixels wide and between 300-600 pixels high.

Submitted.

- There is possible reuse of images between figures in several cases. Please carefully check reuse within the figure set and call out all image reuse in the corresponding figure legends. Possible reuse between:

Figure 1i and Figure 3E

The image reuse is indicated in the legend of Figure 3E.

Figure 1G and Figure 4A

The image reuse is indicated in the legend of Figure 4A.

Figure 1D and Figure EV4B

The image reuse is indicated in the legend of Figure EV4B.

Figure EV4 A, A' and Appendix Figure S5 & S7.

The image reuse was already mentioned in the legend of Appendix Figure S5. The image reuse note was added to the legend of Figure S7.

Figure EV5A and Appendix Figure S2A

The image reuse is indicated in the legend of Appendix Figure S2.

Figure EV5A & B and Appendix Figure S9

The image reuse is indicated in the legend of Appendix Figure S9.

Figure Appendix Figure S2B and Appendix Figure S9

The image reuse is indicated in the legend of Appendix Figure S9.

- Pixelation in Figure EV and Appendix files. Currently, EV is in one merged file. Please provide High res Individual EV Figures and Appendix file.

High-resolution version of EV and Appendix figures are provided. The figures in the combined appendix file have been updated.

- Figure Legends (main + EV):

1. Please note that the box plots need to be defined in terms of minima, maxima, centre, bounds of box and whiskers, and percentile in the legends of figures 3A, B

Done.

2. Please note that information related to n is missing in the legends of figures 3A, B

Done.

3. Please note that the blue arrows are not defined in the legend of figures 1G, G'.

Done.

4. Please note that the dotted borders are not defined in the legend of figures 1B', C'.

Done.

5. Please note that the magenta arrow heads are not defined in the legend of figures 1K.

Done.

6. Please note that the magenta circles are not defined in the legend of figures 2A, B.

Done.

7. Please note that the black and white arrows are not defined in the legend of figures EV1 D, E.

Done.

- "MATERIAL AND METHODS" should be renamed to "Methods"

Done.

- 5 movie files - playing. They should be renamed to Movie EV1-EV2 with the corresponding callouts, and the legends should be removed from Appendix PDF and zipped with each movie file.

Done.

- Sections need to be named and the order should be corrected: Title page - Abstract - Keywords - Introduction - Results - Discussion - Methods - Data Availability - Acknowledgements - Disclosure and Competing Interests Statement - References - Figure Legends - Table(s) - Expanded View Figure Legends.

Done.

Referee #1:

Major points

The revised manuscript has changes in both new experiments and presentation that make it easier to understand. Furthermore, the rebuttal has more explanations about the limitations of the data and how they should be interpreted. The rebuttal to the minor points all have corresponding changes in the manuscript, but not so for the rebuttal to the first two major points. I strongly urge the authors to move some of these explanations to the text, rather than make the readers have to download the review file for the additional insightful discussion. Readers would benefit from these technical explanations because the results achieved in this paper are at the limits of the current technology.

"Our claims should indeed be taken as hypotheses, based on differences of DNA content and core density patterns." -- This statement is in the rebuttal and should also be in the abstract. As worded, the abstract sounds too confident that conformationally variable non-canonical structures were observed. The non-expert reader will take in this statement and the multitude of unaveraged cryo-ET densities + docked atomic models, and then conclude that structures like these exist inside cells. I suggested rewording the abstract as follows:

"Additionally, nucleosome conformational variability with non-canonical structures" →
"Additionally, densities consistent with nucleosome conformational variability with non-canonical structures"

A caveat like this does not detract from the impact of the paper because it provides a more accurate description how one can understand the current state of the art.

We thank the reviewer for this insightful comment. We fully agree with the concern and revised the text accordingly to prevent any possible misinterpretations.

To avoid unconfirmed claims, we have separated the final sentence of the abstract into two distinct statements, each conveying a clear and specific message.

The first reports the nucleosome variability *in situ* confirmed in this study:

"Nucleosome conformations could be identified on individual particles in favorable orientations without structure averaging."

The second refers to the observation of particles containing between one and three DNA gyres, which we cautiously associate with non-octameric nucleosomal particles without making a definitive claim:

“Additionally, we observed particles accommodating ranging from less than one to three DNA gyres, which resemble the previously proposed non-octameric nucleosomal particles that accommodate variable DNA wrapping.”

Fig EV4 caption: "1 gyre sub-nucleosome particles seen in side view in the XY plane (A') are unambiguously characterized by the visualisation of a single DNA gyre"

"When seen in top view in (or close to) the XY plane, they can be recognized by their characteristic histone pattern. In some of them, the absence of the second tetramer appears as missing densities."

-- and --

Fig S5 caption: "The single DNA gyre is unambiguously recognized in all denoised (Warp/nad) and raw tomograms."

These two captions sound overly confident that the densities they pointed out are actually (non-canonical) nucleosomes. For example, I'm not sure if the "raw" panel in Figure S5 can be considered unambiguously a nucleosome (or a sub-nucleosome), let alone being able to count the number of gyres. These figure captions (and their callouts in the main text) can be made more rigorous if "subnucleosomes/nucleosomes" is replaced with "candidate subnucleosomes/nucleosomes" or "subnucleosome-like densities". Again, the implication of uncertainty (taking the claims as a hypothesis) does not detract from the impact of this paper.

We agree that a more cautious terminology would strengthen the clarity and rigor of the manuscript. Accordingly, we have revised the text and figure legends by replacing “subnucleosomes/nucleosomes” with “candidate subnucleosomes/nucleosomes” or “subnucleosome-like densities,” to emphasize the interpretative nature of these assignments.

Minor points

Abstract: Here is my attempt to make one of the phrases less awkward:

"less than one to three gyres" → "ranging from less than one up to three gyres"

We incorporated this suggestion into the abstract.

Fig. 1A - C, 4A: Please flatten the text layers to hide the red spell-check underlines.

Done.

Fig. 1G: I suggest adding the label "simulated" to the figure panel so that people who skip the figure legend don't mistake this image for an experimental projection/class average.

Done.

Fig 1G/G' and P.6: I don't see the M shape. Is the blue arrow suppose to indicate the M-shaped motif?

We thank the reviewer for pointing it out. The reference to the "M-shape" has been removed from both the figure captions and the main text. We updated the figure legends accordingly:

"A representative top view (G) displays an internal density pattern similar to the simulated cryo-EM density of the nucleosome model (G')."

The blue arrow pointing out to the dyad axis of the nucleosome.

Fig EV2: FSC plot -- please add legend.

Done.

Fig EV3 caption: "..indicated by the black arrows". -- Did you mean magenta arrows?

We corrected it.

Fig EV4: "The thickness of the elongated density varies from about 5 to 8.5 nm for 1-gyre particles, and from about 6.5 to 11 for 2-gyres nucleosomes (determined from their histone pattern in XY plane)" -- Is this text referring to the double-arrow-with-dashed-lines in the figure? If so, please indicate this in the text.

Done.

Also, are the solid white arrows in panels A and B' meant to be magenta in color? White arrows are not mentioned in the caption.

We thank the reviewer for pointing out this error. We converted all arrows to magenta.

Dear Prof. Eltsov,

I am pleased to inform you that your manuscript has been accepted for publication in the EMBO Journal.

You may qualify for financial assistance for your publication charges - either via a Springer Nature fully open access agreement or an EMBO initiative. Check your eligibility: <https://link.springer.com/journal/44318/how-to-publish-with-us>

Yours sincerely,

Cornelius Schneider, PhD
Editor
The EMBO Journal
c.schneider@embojournal.org

Please note that it is The EMBO Journal policy for the transcript of the editorial process (containing referee reports and your response letters) to be published as an online supplement to each paper. If you should prefer removal of any referee-only figures included in the point-by-point response(s), e.g. because they may still be used for future publication or because they have been reproduced from published work by others, please do let us know immediately via response email.

More information is available here: <https://link.springer.com/partners/embo-press/editorial-policies#Peer%20review>